# Automatic Extraction of High-Voltage Bundle Subconductors Using Airborne LiDAR Data

**Nosheen Munir** * , **Mohammad Awrangjeb** and **Bela Stantic**

Institute for Integrated and Intelligent Systems, Griffith University, Nathan, QLD 4111, Australia;
m.awrangjeb@griffith.edu.au (M.A.); b.stantic@griffith.edu.au (B.S.)
* Correspondence: nosheen.munir@griffithuni.edu.au

**Abstract:** Overhead high-voltage conductors are the chief components of power lines and their safety has a strong influence on social and daily life. In the recent decade, the airborne laser scanning (ALS) technique has been widely used to capture the three-dimensional (3D) information of power lines and surrounding objects. Most of the existing methods focused on extraction of single conductors or extracted all conductors as one object class by applying machine learning techniques. Nevertheless, power line corridors (PLCs) are built with multi-loop, multi-phase structures (bundle conductors) and exist in intricate environments (e.g., mountains and forests), and thus raise challenges to process ALS data for extraction of individual conductors. This paper proposes an automated method to extract individual subconductors in bundles from complex structure of PLCs using a combined image- and point-based approach. First, the input point cloud data are grouped into 3D voxel grid and PL points and separated from pylon and tree points using the fact that pylons and trees are vertical objects while power lines are non-vertical objects. These pylons are further separated from trees by employing a statistical analysis technique and used to extract span points between two consecutive pylons; then, by using the distribution properties of power lines in each individual span, the bundles located at different height levels are extracted using image-based processing; finally, subconductors in each bundle are detected and extracted by introducing a window that slides over the individual bundle. The orthogonal plane transformation and recursive clustering procedures are exploited in each window position and a point-based processing is conducted iteratively for extraction of complete individual subconductors in each bundle. The feasibility and validity of the proposed method are verified on two Australian sites having bundle conductors in high-voltage transmission lines. Our experiments show that the proposed method achieves a reliable result by extracting the real structure of bundle conductors in power lines with correctness of 100% and 90% in the two test sites, respectively.

**Keywords:** power lines; LiDAR; span; conductors; pylons; extraction

## 1. Introduction

High-voltage power lines are one of the major components of the power transmission system that facilitate the delivery of electricity over long distances with a minimum loss of power [1]. Worldwide, high-voltage power lines have been increased from 5.5 million km in 2014 to 6.8 million km in 2020 [2]. Due to rapid development in transmission network, it is inevitable to avoid mountainous terrains or forests [3] in a power line corridor (PLC). In the long-term, power lines are often infringed upon by harsh weather conditions (e.g., large temperature difference, high humidity, and vegetation encroachment [4,5]), which can intensify flash over discharge leading to large area blackouts, causing significant financial cost and heavy national economic losses [6,7]. Therefore, to guarantee the safe and secure delivery of electricity, it is important to inspect the power lines rapidly

and accurately [8]. The maintenance of PLC is based on two factors: electrical components (e.g., conductors and pylons) and surrounding objects (e.g., trees, shrubs, and vegetation) [9–11].

The conventional methods for inspection of transmission network are in-person (field and airborne) surveys [9]. Although in-person surveys have a high hazard detection rate, this approach is human-dominated, and therefore time-consuming and labor-intensive [9,10]. Furthermore, most of the high-voltage power lines spread to complex terrains with complex distribution (e.g., multi-circuit and multi-bundle) [12], thus make the power lines patrolling and maintenance more difficult and challenging.

Due to an advancement in remote sensing technology in recent decades with continuous development in hardware and innovative data processing algorithms, different modern remote sensing methods (such as videos [1] and optical images [13,14] synthetic aperture radars (SAR) [15], thermal images [16], mobile laser scanning (MLS) [9], and light detection and ranging (LiDAR) [1,7,17]) as well as several other monitoring devices (such as satellite, airborne, and unmanned aerial vehicles (UAVs) [18,19]) have been used for the management of power system infrastructure. A comprehensive review on various PLC surveying methods is given in Matikainen et al. [9].

In comparison to other remote sensing technologies, airborne LiDAR can easily acquire accurate and high density 3D point clouds over a large scene covered with natural and structural objects and the collected data can automatically be processed with built-in powerful computer systems. Moreover, it is highly suitable for forest and hilly terrain due to its access which is not possible with MLS and other vehicle-borne and in-person technologies.

With the continuous advancements in electricity infrastructure, the PLCs are growing and cover undulating terrains (e.g., mountains and forests). (Specifically, in some countries the large part of the PL structure exist in the forest area. Furthermore, natural disasters such as storms, strong winds, bird damage, and vegetation encroachment can cause damage to power lines and requires an immediate need to detect the damage, often in difficult conditions [2,9]. Thus, more robust methods are required for automated and accurate solution for the power lines mapping and monitoring.

In addition, to suppress corona discharge and reactance or to save land occupation, transmission towers are widely constructed with bundle conductors instead of single conductors. Thus, the distribution of power lines has become more complex, with different layers and the increasing use of multi-loops and circuits. Another obvious problem with power lines structure is that the conductors are thin and long, thus the actual number of PL points are far smaller than the number of input points. Furthermore, there could be a long gap along the conductors where there are no points in the input data. These issues raise challenges to process LiDAR data for individual bundle subconductor extraction.

This paper addresses the above challenges by proposing a novel method that first extracts the desired regions in the form of spans containing the single as well as the bundle subconductors. Then, each span is processed individually to extract the individual bundle conductors. The benefits of extracting and processing each span individually are a significant reduction in the size of the input data and improved process efficiency. Moreover, the proposed method does not use any machine learning technique, thus does not require any training data, which is hard to achieve for a large area. In addition, the proposed algorithm is tested on low point density data sets with flat and hilly terrains, thus making it feasible for real PL infrastructure.

Extraction of individual subconductors from bundles is generally a less common research area; most of the existing methods extract PL points as a class or consider bundle conductors as a single conductor for power lines extraction and modeling [20]. It is worth noting that the accurate modeling of PL is highly dependent on the correct extraction of each subconductor. Identification and extraction of the bundle conductors is the primary requirement for precise modeling and mapping of each PL. In the case of maintenance, the mapping of individual subconductors will help localize the faulty conductors. Knowing their locations and properties will help in better assessment management of the utility companies.

In spite of its importance and demands in PLC mapping and modeling, there has been not much research effort that is directly related to individual bundle conductor extraction in PLC using airborne LiDAR data.

The rest of the paper is organized as follows. Section 2 reviews the state-of-the-art methods for the PL extraction using point cloud data. Section 3 introduces the proposed method for bundle subconductor extraction. The details of the data sets and parameters are discussed in Sections 4.1 and 4.2, respectively. Sections 4.3 and 4.4 outline ground truth details and evaluation metrics to verify the proposed method. The results on the test data sets are provided in Section 5. Section 6 provides the detailed discussion on results including the comparisons with existing approaches. Finally, conclusions drawn from experiments are presented in Section 7.

## 2. Related Work

As an imperative application of LiDAR technology, PL inspection has drawn a lot of attention and many studies have been reported on this topic in the last two decades. Generally, based on the outcome and final results, most of the conducted studies on PL extraction are divided into three groups: classification of PL points, single PL extraction in spans, and bundle conductors extraction.

PL points classification: In this category, studies that classified PL points using machine and non-machine learning techniques are discussed. For example, in a very early work on PL points classification, McLaughlin [21] computed the Gaussian mixture model using the expectation minimization algorithm and clustered the point cloud data into three clusters: transmission lines, vegetation, and surfaces (e.g., buildings) using an elliptical neighborhood. Clode and Rottensteiner [22] transferred the height differences between the first and last pulses and the laser intensity values into a grid space and then classified the grid data into trees and PLs using the Dempster–Shafer classification model. Due to the absence of the ground truth, only visual results were presented for PL classification. Melzer and Briese [23] first applied an iterative HT on the grid data to extract lines, and then clustered the lines to obtain the initial position of pylons. Finally, points within successive pylons were classified to reconstruct PLs using the RANSAC algorithm, where the initial hypotheses were obtained by a Neural Gas Network. Sohn et al. [24] used a Markov Random Field (MOV) model to classify LiDAR data into PL, pylon, and building points using linear and planar features. Based on the classification results, pylon locations were detected and spans were identified. Finally, power lines were modeled with catenary curve models in 3D.

Kim and Sohn [25] introduced a 3D point-based supervised classification method using the Random Forest (RF) classifier and divided the objects in the PLC into five classes i.e., power lines, pylons, vegetation, buildings, and ground. Guo et al. [26] proposed another classification method by first extracting 26 different features using three types of neighborhoods (cylindrical, sphere, and grid) and then applied the Joint Boost classifier to classify the objects into ground, vegetation, building, pylon and power lines classes. In some recent work, Wang et al. [27] proposed a semi-automated PL classification method by first identifying the PLC direction using the Hough transform (HT) with the RANSAC (random sample consensus) algorithm and later applied the Support Vector Machine (SVM) classifier to classify PL points using a slanted cylindrical neighborhood. All of these methods focused on classification of PL points as one object or class, instead of extracting individual power lines.

Single PL extraction: Most of the existing studies in the literature focused on extraction of single power lines. For example, Liu et al. [28] proposed a statistical method where the data points on a single PL were extracted based on the fact that points on the same PL were much closer as compared to the points on different power lines. However, this is not always true, particularly for thin conductors when there are long gaps exist between the successive points on the same conductor. Zhu and Hyyppä [29] used a statistical analysis involving height, density, and histogram thresholds for candidate point selection, and then generated a 2D grid with the candidate points for a continuity and shape-based analysis to separate power lines from other objects. They applied a set of thresholds on height, density, and histogram that may result in poor adaptability with different data sets. Cheng et al. [30]

presented a voxel-based hierarchical method integrating single voxel filtering with neighboring voxel filtering for extracting power lines based on the spatial distribution characteristics that conductor points are closely linked on the same PL.

In some recent works, Yadav and Chousalkar [31] proposed a HT-based technique to extract the power lines from vehicle-borne LiDAR point clouds. Guan et al. [32] used vehicle-borne LiDAR Data to detect power lines by analyzing the 3D morphology. Qin et al. [33] extracted the power lines using a cable inspection robot (CIR) technique in which the CIR travels over conductors and rapidly extracts PL point by position and orientation system (POS) extraction model. They later processed the points to remove the noise using voxelization technique. Yang et al. [10] developed a voxel-based method for PL extraction with the Markov random field model, and Laplacian smoothing was applied to get the skeleton structure of power lines, and finally latent Dirichlet location topic models were used for voxel feature construction and PL extraction.

In a very a latest work, Jung et al. [34] proposed a two-step process based on extraction and refinement of candidate PL points. In the candidate PL point extraction step, input point cloud data was subsampled using a 3D voxel grid that preserved the 3D details of point cloud but significantly reduced the data set size. Then, ground filtering was performed to get ground elevation and height filtering was used to remove unwanted objects within a certain height exist above the ground. After removing the ground surface and other objects, candidate PL points were extracted by removing the unwanted objects near the power lines. In the refinement step, the candidate PL points were recovered in original form and processed further by applying image- and cluster-based techniques. By fitting mathematical models, an individual PL was re-clustered and used to reconstruct the broken sections in the power lines. The proposed method achieved the total precision and recall rates of 93.39–96.76% and 82.58–97.65%, respectively, over 30 diverse data sets acquired in four different sites. However, the results provided were on single PL spans and bundle subconductors were not extracted.

Bundle conductors extraction: Extraction of bundle wires is still a challenging task; as discussed above, most of the studies paid attention to the extraction of single PLs. For example, Jwa et al. [35] proposed a perceptual grouping framework based on Gestalt laws, which could extract relevant groupings and structures from organized points, fragments, and segmenting elements with similar characteristics for detecting the power line points. The compass line filter later used to detect the power line points that belong to each line. The author claims to extract up to two bundle conductors, but the results are not visually shown. In some latest work, Zhou et al. [20] first classified the input points into four classes: ground, vegetation, power lines, and pylons using the Joint Boost classifier. The single power line spans were then extracted by a spatial clustering-based method and bundle conductors were identified by analyzing the fitting residual of single power lines. Finally, the bundle conductors were extracted by projecting them on the XOZ and XOY planes and separated using a least square based line fitting technique. Finally, a double-RANSAC algorithm was used to reconstruct the extracted conductors. The proposed method was tested on different spans that contained bundle conductors and showed a good accuracy. However, the proposed method is not fully automated and, thus, requires the training data set containing PL points.

Awrangjeb [36] proposed a hierarchical approach where PLCs, pylons, and power lines were extracted in order. The PL corridors in the form of straight lines were first extracted by converting the input point cloud data at different height levels into binary images. The series of convex hulls were formed around the straight lines and projected on the horizontal plane at each height level to get stand alone corridors. Each extracted PLC consisted of a set of rectangular regions that connected serially with each other to form a polygon which defined the parameters of the PLC. Then, only points within each rectangular PLC region were considered to locate and extract pylons. Finally, the non-ground points between two successive pylons of the same PLC were used to extract individual power lines. For extraction of individual conductors between any two successive pylons vertical and horizontal masks were generated. The cluster of power lines at different height levels were identified firstly by vertical mask, and then horizontal mask was used to count the number of individual conductors in

each cluster. Finally, for each PL a seed region is defined and, then, extended on both sides to extract the whole PL. The method was tested on two big data sets showing high point- and object-based accuracies, though the proposed method was capable to extract maximum two subconductors from each bundle.

Munir et al. [37] proposed a semiautomated hierarchical method where the SVM was first used to separate the PL points from other objects. Then, by exploiting the spatial distribution property of power lines, a voxel-based technique was applied to separate individual conductors. Results were presented with high accuracy on two large data sets, but the method only extracted a maximum of two subconductors from each bundle. In another work of Munir et al. [38], bundles containing a maximum of four conductors were extracted. First, the bundles located at different height levels were detected in the form of clusters by using the fact that conductors in one bundle were close while compared to the conductors existed in a different bundle and a conductor count was generated on each cluster. The cluster points were projected on a perpendicular plane which is orthogonal to the direction of the span. Finally, individual conductors were separated by fitting the 2D line and by using the point and distance formula. Experimental results were shown on different spans from two different data sets. The method was capable of extracting bundle conductors up to four with high completeness and correctness values, although the method was dependent on some strict parameters from the data sets such as pylon contextual information.

## 3. Methodology

This paper presents an automated method for extraction of individual conductors that exist in a bundle by combining grid (i.e., 3D point cloud data is interpolated into a 2D grid space where each pixel represents a point in the input 3D data) and point-based (i.e., every single point is considered in its 3D form for carrying full investigation of input data) approaches in order to leverage the benefits from both. However, the method does not use any classifiers to avoid the shortcomings related to machine learning techniques.

As stated above, most of the existing studies focused on single PL extraction in a span, or they considered bundle conductors as a single PL, while limited studies have been reported for extraction of multiple conductors in bundles. This paper exploits the distribution characteristics of bundle subconductors in high-voltage PLC, and offers a novel method for their extraction. The proposed method can robustly and precisely extract each individual conductors from the bundle.

The proposed approach first extracts the PL candidate regions in the form of spans by adopting the technique mentioned in Munir et al. [37]. Once the spans are determined, the main contribution of this paper comes, which is the extraction of subconductors from each bundle located in the span. These subconductors are actually several parallel cables located at intervals by spacers in each bundle. Figure 1 shows the flow diagram of the proposed method. First of all, the span and pylon locations which determine the starting and ending of points in each span in a PLC are extracted. Then, each span is converted into a 2D binary mask for extraction of bundles. A sliding window is formed on each bundle that slides iteratively over the bundle. The bundle points located within the window are projected on a plane which is perpendicular to power lines direction. These projected points are separated into clusters which determines the number of conductors in a given bundle segment. This window is moved iteratively towards the other end, when in each iteration the new window position overlaps the previous window, thus contain some new points and new clusters of conductors are generated. These new clusters are merged with the previously generated clusters by applying the AND operation among them. Consequently, each subconductor from the bundle is extracted until the window reaches the other end of the span.

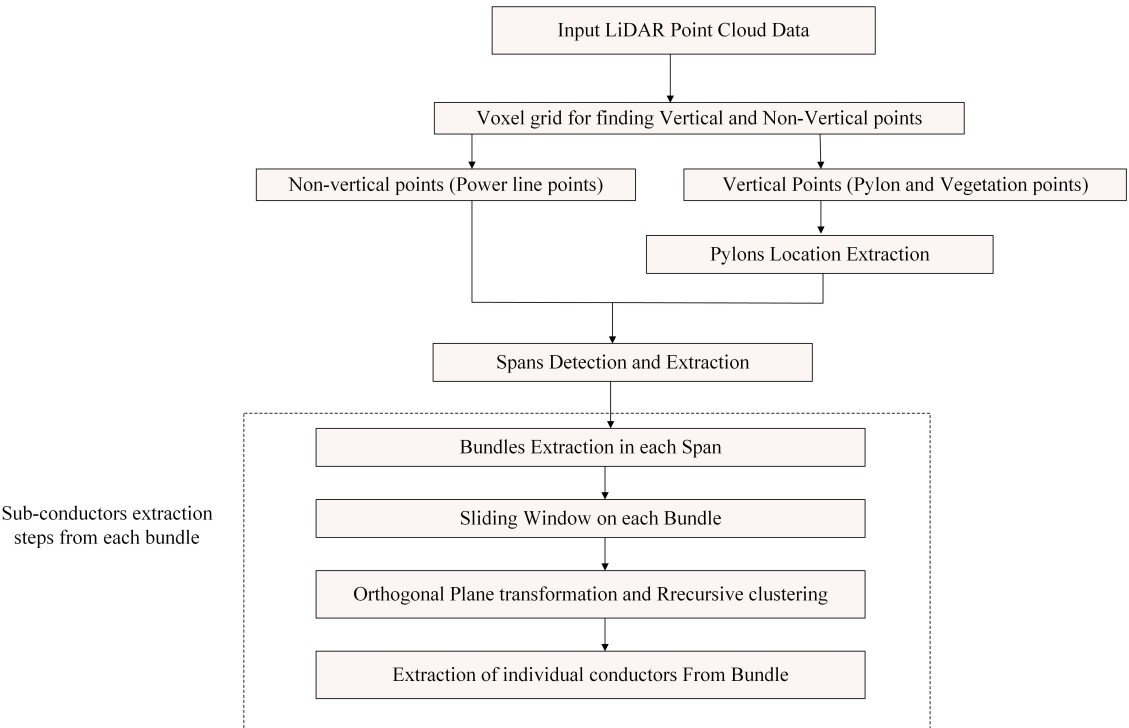

**Figure 1.** Processing flowchart of the proposed method.

First of all, the method for detection and extraction of pylons and spans is simply introduced in Section 3.1. Then, the steps for detection and extraction of bundles are explained in Section 3.2. Finally, the extraction of subconductors in each bundle is discussed in Section 3.3.

### 3.1. Extraction of Span and Pylon Locations

To decrease the size of 3D point cloud data for subsequent steps, it is important to locate the candidate regions of each PLC first, and then to extract the PL points in the detected candidate regions. A PLC consists of multiple spans. A region between two successive pylons is termed as a span, and to extract these candidate regions it is mandatory to detect the pylon locations first. To detect and extract these regions, points within the height of 1 m above the ground are removed and the remaining (non-ground) points are separated into two classes, i.e., a vertical class that contains the pylons and a vegetation and non-vertical class that contains the PL points only, by exploiting the fact that power lines show vertical discontinuity as they are hanged above the ground at certain height and detached from each other as well from the ground, while pylon and vegetation show vertical continuity [39]. To separate these classes, a 3D voxel grid of size of 5 m × 5 m × 5 m is created over the input data to preserve the geometrical information of points in each grid cell (voxel). For separation of vertical and non-vertical points, each cuboid in a grid is further divided into fixed sections of 1 m height. The number of "continuous on segments" $C_n$, i.e, the consecutive segments with points, and the number of continuous off segments' $C_f$, i.e, the consecutive segments with no points for each voxel, are calculated. While, vegetation and pylons show high value for $C_n$ but power lines display the opposite. Finally, pylons are separated from the trees by employing the statistical analysis technique presented in Munir et al. [37].

As mentioned earlier the transmission lines in any PLC is connected in the form of spans. These spans are connected to each other through pylons. Once pylon locations are determined, they are further used to get the span points, i.e., PL points between two consecutive pylons. This approach will help us to concentrate on the input points within the spans. Each span is processed individually, thus it makes the further processing of data easier and reduces the total amount of data that needs to

be analyzed for extraction purpose. It is important to note here, only the non-ground points extracted in this step are considered as span points and will be used for further processing in the next step.

### 3.2. Extraction of Bundles

For extraction of bundles, the understanding of PL design and its structure characteristics is paramount. Generally, these characteristics come from the construction specifications of pylons (tower): power lines must have adequate spacing between them, they can not intersect each other, they are hung above the ground at different height levels, and they appear as straight lines when projected on a horizontal plane. Thus, it is important to understand the structure of pylons with multi-circuit and multiphase structure prior to bundle extraction. A typical double circuit two-bundle PL tower is shown in Figure 2. For this tower, all power lines are divided into different transmission circuits (Circuit I and Circuit II) according to their distribution in the tower structure. The tower has cross-arms at four different height levels and the two conductors attached to the cross-arms at each level of a transmission circuit are considered as a bundle. Therefore, there are total of six bundles ($6 \times 2$) hanged from the cross-arms. In this section, we intend to find the number of transmission circuits to extract points in each transmission circuit and then the extraction of individual bundles from each circuit.

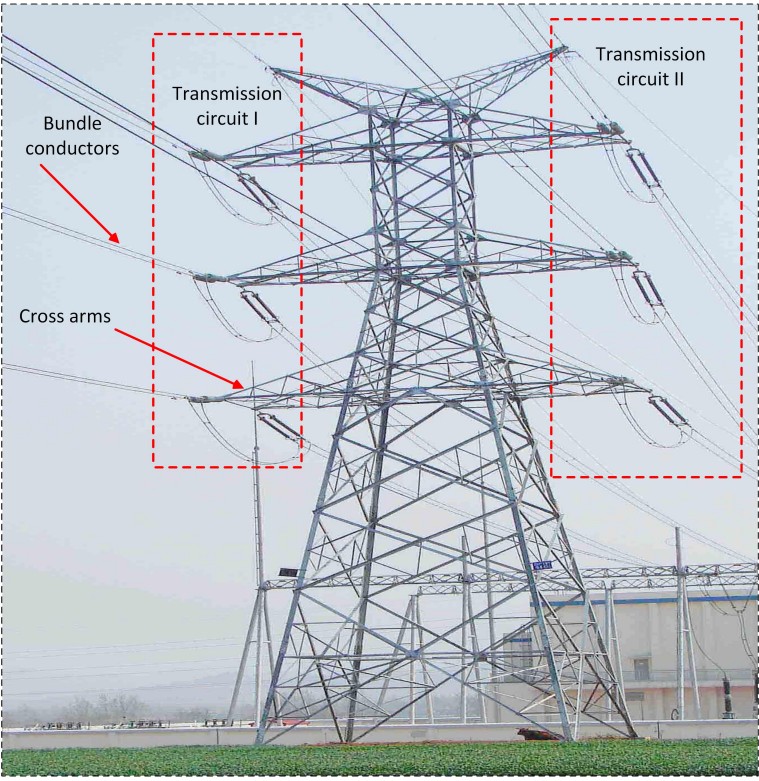

**Figure 2.** A typical transmission line tower. Source: Adapted from the work in [40].

To achieve this task, firstly a binary mask $M$ with the points within each span is generated following the process in Awrangjeb et al. [41]. The resolution of the mask is set fixed at 0.25 m and all pixels are initially filled with 0 (zero). Then, for each non-ground point within a pixel, a neighborhood (e.g., $3 \times 3$, consistent with the point density) is filled with 1 (white). In $M$ all the points in each transmission circuit construct a thick straight line.

Figure 3 shows two spans in 3D (Figure 3a,f) and 2D (Figure 3b,g, after projection on a horizontal plane) views with their corresponding binary masks in Figure 3c,h, respectively, from two different test data sets. It can be observed that the span in Figure 3a has two transmission circuits (double-circuit) and that in Figure 3f has three transmission circuits (triple-circuit). The bundles in each transmission

circuit at different height levels in the 3D view overlapped in the 2D view, so they form thick straight lines due to absence of height factor.

Therefore, we can easily count the number of transmission circuits as well as extract the points in each circuit $C_p$. Because the PL points that belong to a overlapped bundle are much closer to one another as compared to the PL points that belong to a different overlapped bundle. By exploiting this fact $M$ is flood filled so that the black holes inside each overlapped bundle are filled with white. This can avoid the issue raised because of low point density or absence of data. Consequently, the PL points that belong to a overlapped bundle are connected as shown in Figure 3c,h. Finally, a connected component analysis (CCA) is carried out on the filled image to count the number of transmission circuits and to obtain their individual boundaries. Each of the connected component (CC) contains the set of points $C_p$ of the related transmission circuit. Figure 3d,i shows the connected components for the transmission circuits in each span. Figure 3e,j shows the $C_p$ within two selected circuits, respectively.

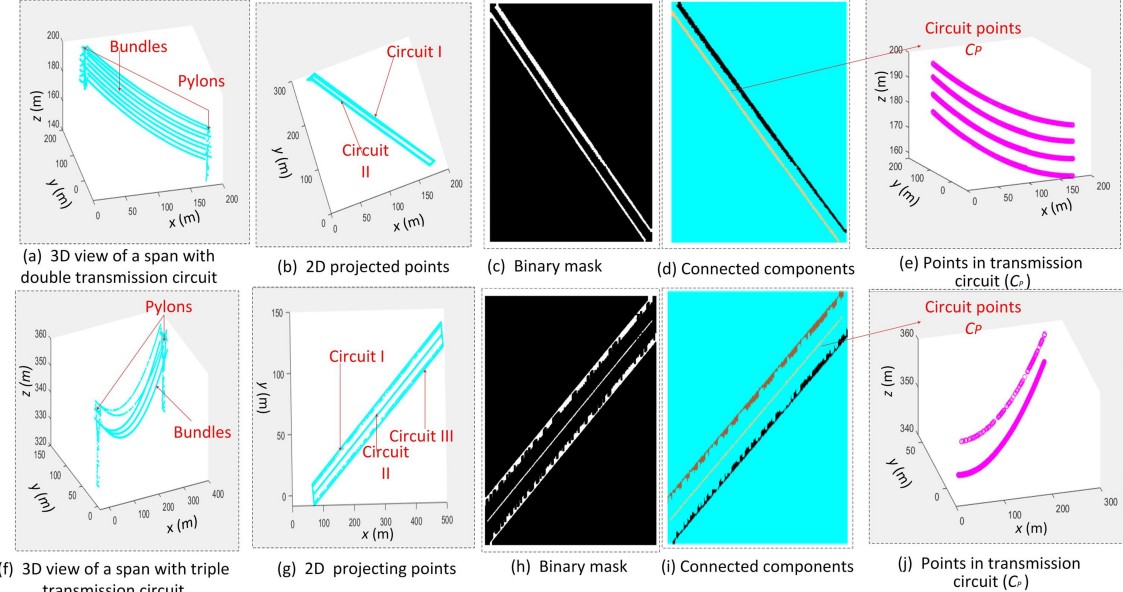

**Figure 3.** Extraction of the set of points in each transmission circuit of a double-circuit span (**a**–**e**) and a triple-circuit span (**f**–**j**). In panels (**a**–**e**) the origin (0, 0) represents ($5.1435 \times 10^5$, $6.9225 \times 10^6$) and in panels (**f**–**j**) the origin (0, 0) represents ($4.08000 \times 10^6$, $5.90355 \times 10^5$).

The points in $C_p$ are divided into several segments, each 12 m long along the span direction. Figure 4b shows the segments (separated by cyan dash lines) of $C_p$ shown in Figure 4a. The length of each segment should be small enough so that each bundle in the segment should appear as a straight line (see Figure 4c). It should be noted that due to the catenary curve nature of power lines in a span, i.e., the PL height is low in the middle of the span and high at the two ends, thus the height of PL points may vary along the span direction (see Figure 3f).

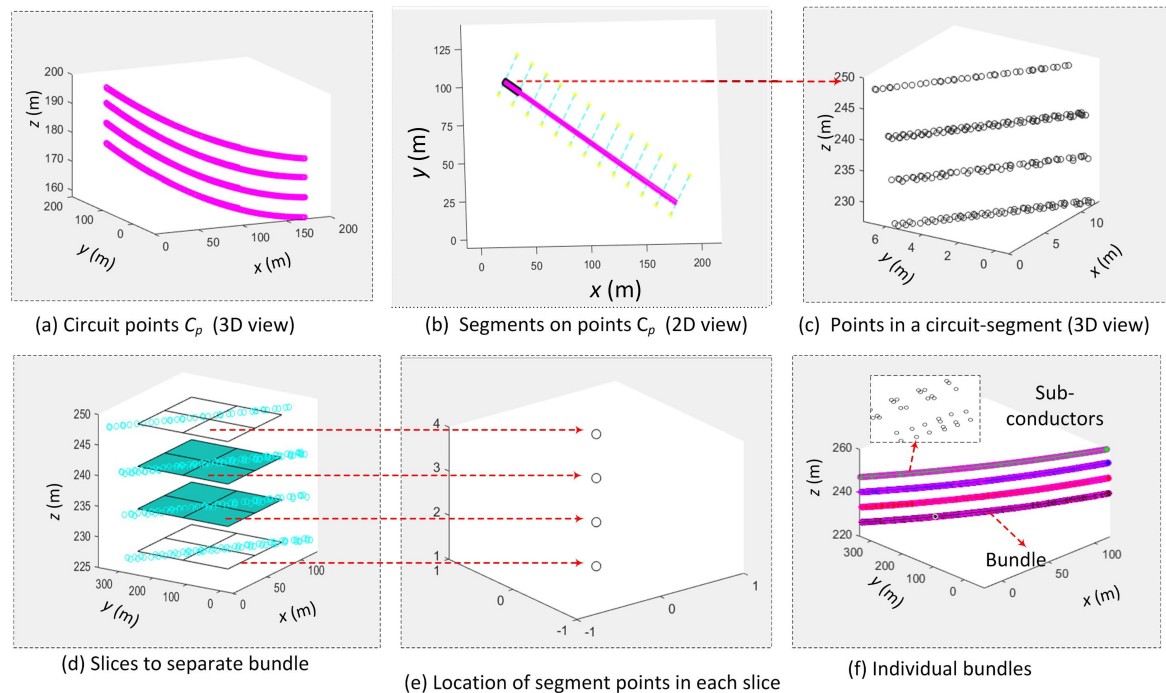

**Figure 4.** Detection and extraction of bundle points. In panels (**a–d,f**) the origin (0, 0) represents $(5.155 \times 10^5, 6.92176 \times 10^6)$.

Each circuit segment is expected to have bundle conductors at different height levels. This distribution property is exploited to form 3D vertical slices based on the minimum and maximum of heights ($z$) of the segment points (Figure 4d). The number of slices $s_n$ is empirically set to 4 by considering the fact that pylons or towers in the given data set are built with maximum of four cross-arms. This value can be increased if the pylon is built with more cross-arms. The points in the segment can be located in one of the slices depending on their heights. Thus, according to their location slice numbers are allocated as shown in Figure 4e. All the points in a bundle within the segment get the same slice number. If the points of a bundle in the segment are located between two slices, the bottom slice number is allocated to them, or if they are intersecting any slice then the corresponding slice number is assigned to them. Finally, the bundle points with the same slice number are merged to have the individual bundle points $B_p$ for the whole span (Figure 4f. If a bundle consists of more than one conductor, these conductors are called subconductors. Figure 4f shows each bundle in different color and in the magnified view the subconductors are shown for the top bundle. In the following step, points $B_p$ in each bundle will be processed for extraction of individual subconductors.

### 3.3. Extraction of Individual Conductors

In order to extract the individual conductors (subconductors) from the points $B_p$ of a bundle, a local bundle coordinate system ($X_b$, $Y_b$, $Z_b$) is defined, where the origin $O_b$ represents the minimum of $x$, $y$, and $z$ of points $B_p$ (see Figure 5a). The $X_b$-axis and $Z_b$-axis are parallel to the directions of the span and the height, respectively. A 2D square window of size $4 \times 4$ m$^2$ is then placed on the starting position of the bundle. The window length (along the $X_b$-axis) is set by considering the fact that points from $B_p$ within the window appear in straight lines. Figure 5b shows the points within the first window position in Figure 5a. This window is moved forward in the direction of $X_b$-axis iteratively. For extraction of individual conductor points in a given window position, a projected dichotomy in connection with distance based clustering is introduced. To start this procedure, points within each window position are projected onto the 3D orthogonal plane, shown in Figure 5a, where the plane is perpendicular to the direction of the window (the $X_b$-axis). Figure 5d,e shows the projected points in 3D and 2D views, respectively, on the orthogonal plane. These translated points are further separated

into individual conductors in the form of clusters by using their $x_b$ and $y_b$ coordinates. The clustering is performed using the distance between the projected points. Each point is clustered with its closest neighboring point if the distance between the two points is shorter than a predefined threshold. Since, the minimum distance between two conductors in a bundle is 0.25 m, this threshold is used to separate individual conductor points from the given window. Each cluster in Figure 5e represents the points of a subconductor in the bundle. These clusters are converted back to their original 3D coordinate system (see Figure 5f) and they will be used in the next iteration.

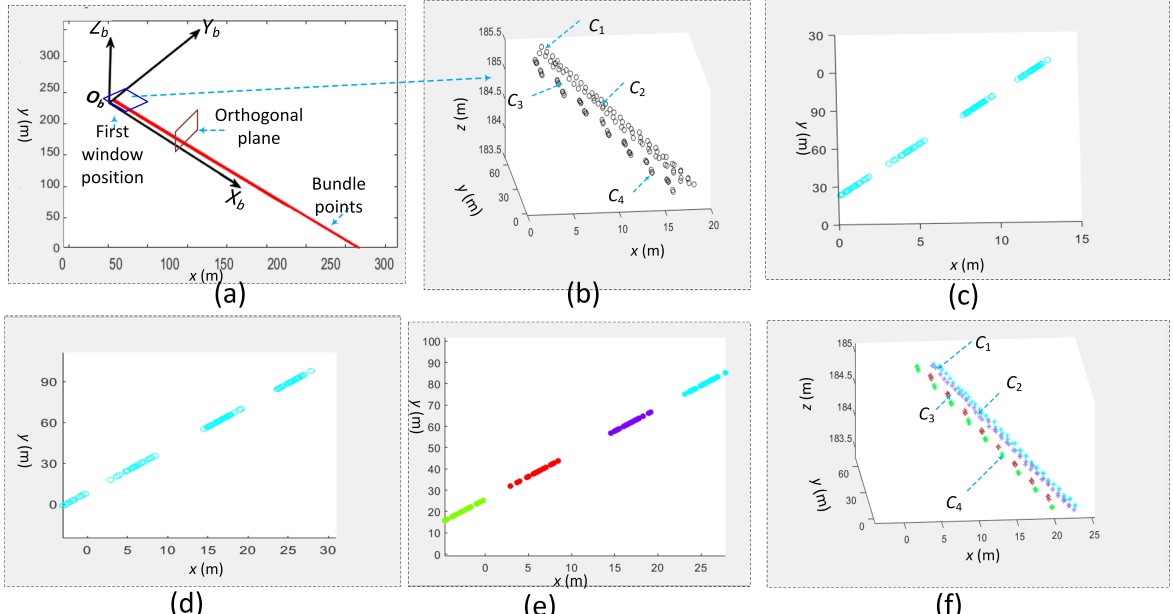

**Figure 5.** Subconductor extraction using a sliding window: (**a**) local coordinate system, sliding window, and orthogonal plane; (**b**) Conductor points within the first window position; (**c**) 3D view of projected points on orthogonal plane; (**d**) 2D view of projected points on orthogonal plane; (**e**) 2D view of projected point clusters on orthogonal plane; (**f**) individual conductor points (clusters) in different colors. In panels (**a**,**b**,**f**) the origin (0, 0) represents $(5.1436 \times 10^5, 6.9226 \times 10^6)$ and in panels (**c**–**e**) the origin (0, 0) represents $(5.14398 \times 10^5, 6.9247 \times 10^6)$.

Figure 6a–c shows the two clustered conductors where a spacer between them did not affect the outcome except the spacer points are misclassified as conductor points. After the first iteration, the sliding window is now moved 1 m forward iteratively towards the span direction. Figure 6d shows the successive window positions, while the width (along the span direction) of the window is set to remain the same at 4 m but its length (across the span direction) is increased to 1 m, i.e., 0.5 m on each side. This change in the window size takes care of the cases when the conductors may change their directions within the span due to their swing behavior under windy conditions or if errors associated with the LiDAR data acquisition are present. We need to make sure no bundle points are missed in each iteration due to the unexpected swing or data acquisition error in the span.

Moreover, the gradual approach (advancing the window 1 m at a time) helps connect the points from the same conductor, even though there is a gap (no points) along the conductor. Between any two successive window positions there is a 3 m overlapping area. Points from this area will connect the points from the two non-overlapping areas at the two ends of the successive window positions. Consequently, if there is at least one point within a 3 m distance along a conductor the conductor points can be extracted effectively.

The two steps mentioned above, i.e, projection on the orthogonal plane and distance-based clustering, are performed at every window position. Thus, the individual conductor points are updated and stored in the form of clusters in each iteration. After the iterative procedure, the total

number of clusters corresponds to the number of extracted subconductors from the given bundle points. Ideally, PL points with no breakage (gaps shorter than 3 m, Figure 6) and high point density in each window position result in correct extraction of individual conductors and smooth update in each iteration.

However, in practice if there are breakages (gaps longer than 3 m) between two successive points on the same conductor, that conductor can be extracted as more than one conductor. This can happen due to low density input data or a problem during the data acquisition. In this case, we can have more clusters as compared to the actual number of subconductors in the bundle.

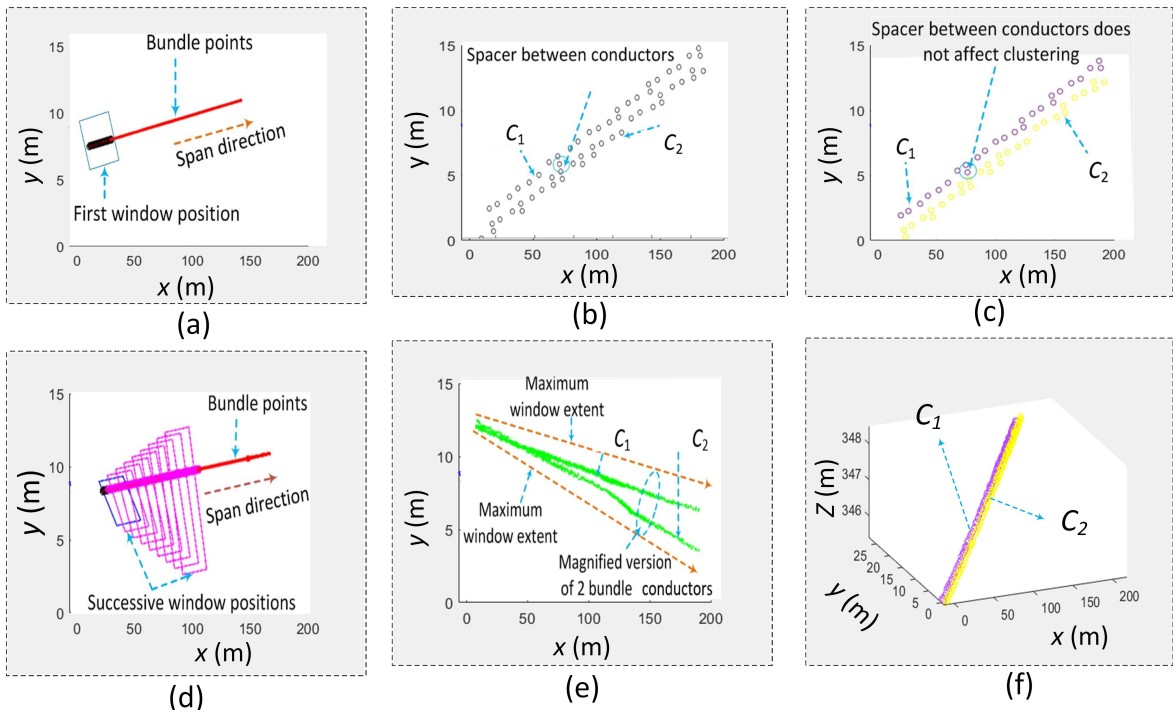

**Figure 6.** Complete subconductor extraction from a bundle: (**a**) initial window position; (**b**) points within the initial window; (**c**) two clustered conductors; (**d**) successive positions of the sliding window; (**e**) magnified version of part of the bundle; (**f**) final clustered conductors. In panels (**a**–**f**) the origin (0, 0) represents $(4.0808 \times 10^5, 5.90348 \times 10^6)$.

Thus, there is a need of another step, which can circumvent this issue and merge the clusters which belong to the same conductors. The mean point of each cluster in each window position is estimated. Then, the distance of each cluster in each window position is checked with the means of the clusters in neighboring window positions after transforming them on the orthogonal plane. If any two clusters belong to the same conductor, their mean points will have a distance less than or equal to 0.15 m. This value is less then the minimum distance value (0.25 m) used earlier for separation of conductors in a bundle. However, the error can still occur if the gap is too long, when the clusters cannot be merged.

## 4. Performance Study

In this section, the test data sets, ground truth data, parameter settings, results, and discussion are presented in detail.

### 4.1. Data Sets

To test the feasibility of the proposed method, two Australian data sets depicted in Figure 7a,b from Maindample (MDP), Victoria and Bindebango (BDB), Queensland with point density of 23.7 and

56.4 points per m$^2$, respectively, are used. There are three main objects in these data sets: vegetation, pylons, and power lines. Inevitably, noise may exist due to some external factors such as wind, scanner during working, and vibration. The MDP has only two transmission line corridor (TLCs) and one distribution line corridor (DLC). The only distribution line corridor is under the two TLCs is just 300 m long and does not have enough points. While the BDB data set has three TLCs.

The length of the MDP site is about twice the length of the BDB site, hence the point cloud of the BDB site increased proportionally. The width of the MDP site remains constant to 330 m for each corridor in the data set and located on a flat terrain as shown in Figure 7a. While the BDB site is from a hilly terrain and its width changes from 330 m and 530 m across the TLCs, which means spans in these corridors changes their directions considerably (see Figure 7b).

Tables 1 and 2 summarize their properties. In each table, "*n*-conductor bundles" represents the number of bundles that have *n* subconductors in each. The last column represents the total number of conductors exist in each corridor.

Manual inspection of individual spans in the two data sets has shown that the number of power lines and their structure may change even in the same corridor. For example, in Corridor 1 of the BDB site, some span has a total of 14 conductors, while other span has a total of 28 conductors. Figure 8a shows Span 5 in Corridor 1 has a total of 28 conductors in 8 bundles, i.e., 2 bundles at each of 4 height levels and each bundle has 4 conductors, except the top two bundles which contain 2 conductors each.

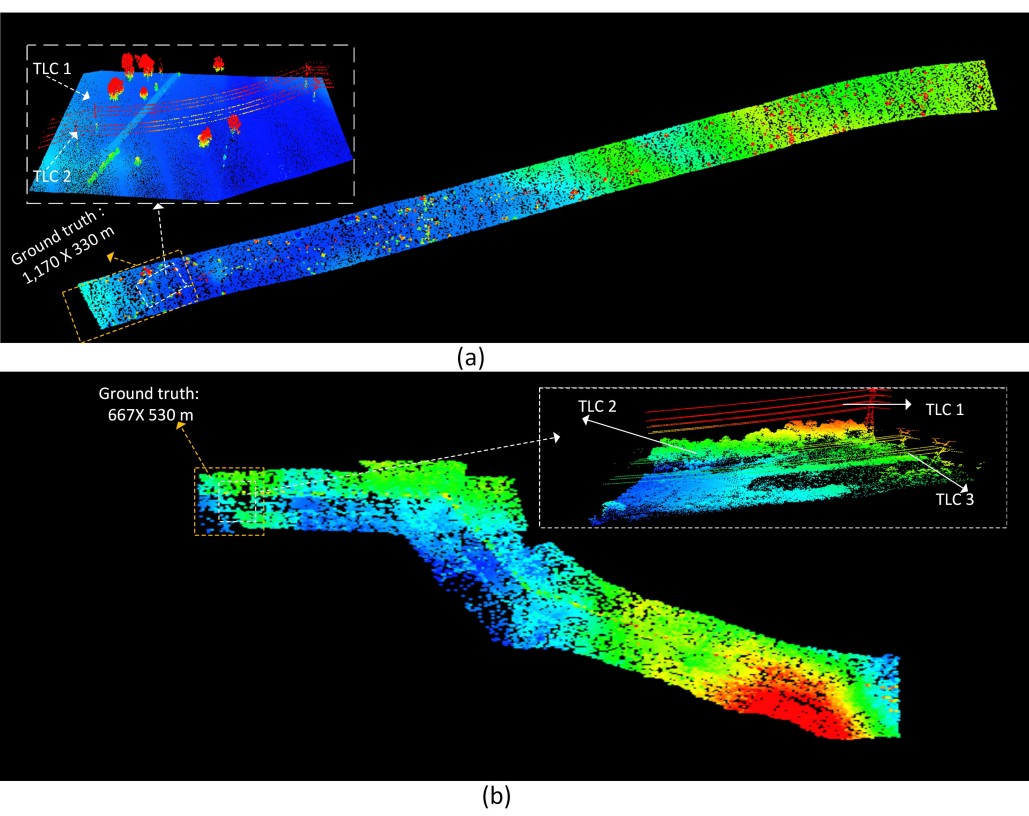

**Figure 7.** Test data sets: (**a**) Maindample (MDP) and (**b**) Bindebango (BDB).

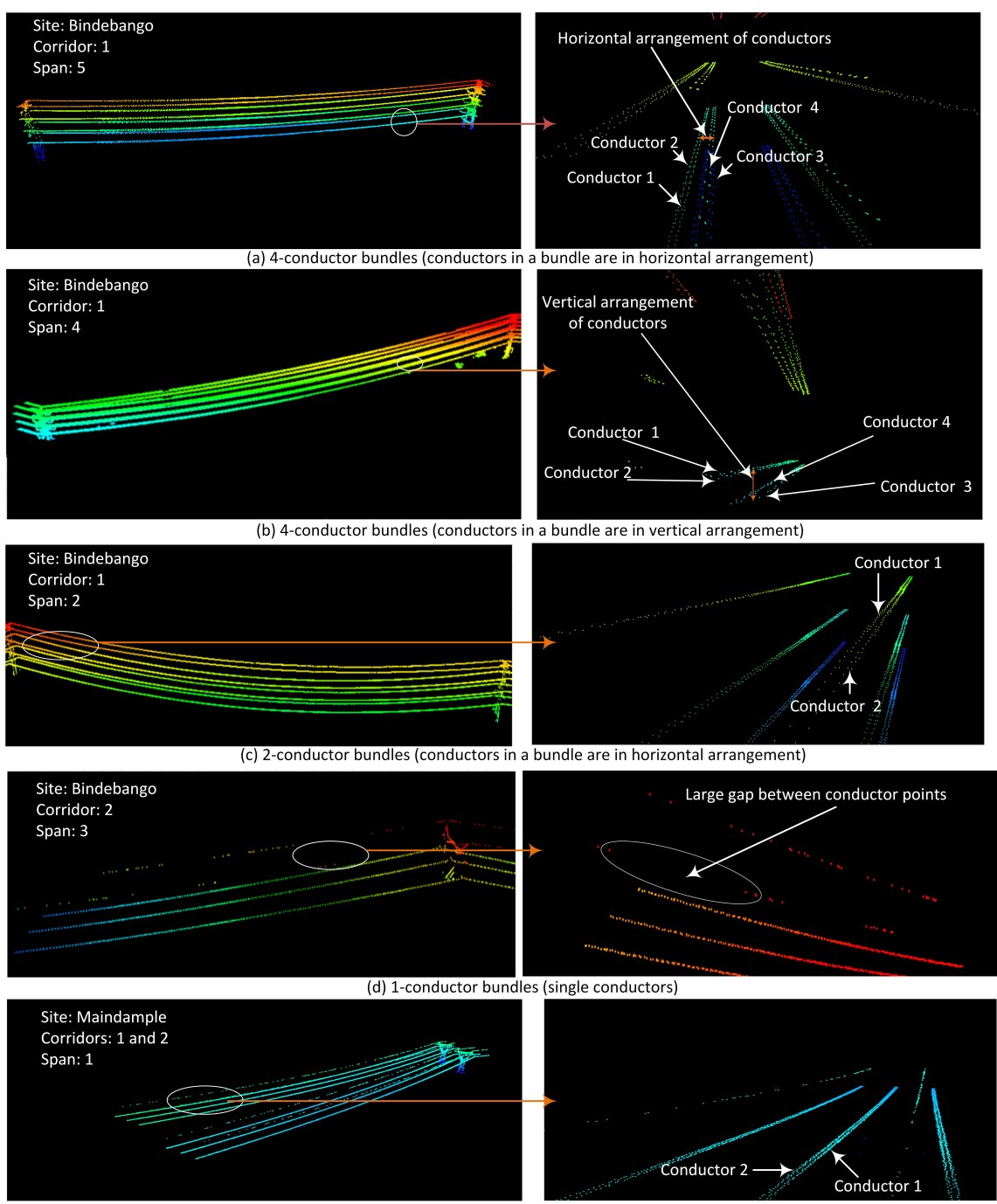

**Figure 8.** 3D view of spans from Bindebango and Maindample sites.

**Table 1.** Summary of Maindample (MDP) data set.

| Corridors | Areas (m²) | Spans | Pylons | 2-Conductor Bundles | 1-Conductor Bundles | Total Conductors |
|---|---|---|---|---|---|---|
| 1 | 5460 × 20 | 14 | 13 | 42 | 28 | 112 |
| 2 | 5460 × 20 | 14 | 13 | 42 | 28 | 112 |
| 3 | 310 × 5.5 | 2 | 3 | 6 | 6 | 18 |
| Total | | 30 | 29 | 90 | 62 | 242 |

**Table 2.** Summary of Bindebango (BDB) data set.

| Corridors | Areas (m$^2$) | Spans | Pylons | 4-Conductor Bundles | 2-Conductor Bundles | 1-Conductor Bundles | Total Number of Conductors |
|---|---|---|---|---|---|---|---|
| 1 | 3000 × 12 | 10 | 8 | 42 | 12 | 24 | 216 |
| 2 | 3000 × 18 | 10 | 8 | 2 | 21 | 15 | 65 |
| 3 | 3000 × 18 | 10 | 8 | 1 | 25 | 17 | 71 |
| Total | | 30 | 24 | 45 | 58 | 56 | 352 |

The conductors in each bundle are in horizontal arrangement. However, Figure 8a shows Span 5 in the same corridor although has the same number of bundles and conductors, the conductors in each bundle are in vertical arrangement. Figure 8c shows Span 2 in the same corridor has 2 conductors in each bundle at three different height levels and 2 single conductors on the top (4 + 4 + 4 + 2). In each of Corridors 2 and 3 of the BDB site, most of the spans have five single conductors at two height levels (3 + 2) (see Figure 8d). The point density in these corridors is quite low and the points in the same conductor have large gaps which makes the individual conductor extraction very challenging. Figure 8e shows the two spans from Corridors 1 and 2 in the MDP site. The number of conductors in the MDP site remains constant throughout each corridor. Each span has a total of 8 conductors at two height levels (6 + 2). There are total 13 spans in each corridor and each corridor is built with three circuits.

*4.2. Parameters*

Table 3 displays the list of parameters with their values used in the proposed method. Figure 9 shows some of these parameters on a sample scene from the Maindample data set. All of these parameters have been tested on the both data sets. Most of the parameters have been empirically set by observing the general infrastructure of the TLCs. For example, any two circuits cannot intersect each other and they must have a good space between them. Most of the parameters, such as the segment size and the initial window size, are chosen by adopting a trial and error method, and it is concluded that these parameter values should not be too big as conductors will not appear as straight lines in the given segment due to sag. Similarly, the size of the window can be increased if the point density is high in the data sets to reduce the iterations and complexity. The other parameters such as the maximum distance between clusters and the distance between mean points of the clusters are sensitive to the approach and have been set by using the fact that the points belonging to a conductor are close to one another as compared to the points belonging to a different conductor. Therefore, it should be based on the minimum distance between two conductors and should not be large, otherwise it will merge the clusters of two different power lines.

**Table 3.** Parameter values used in the proposed approach.

| Sections | Parameters | Values | Sources |
|---|---|---|---|
| Section 3.1 | Ground points removal height | 1 m | [37] |
| Section 3.1 | Voxel (cube) side | 5 m | [25] |
| Section 3.1 | Mask resolution | 0.25 m | [37] |
| Section 3.2 | Distance between circuits | 10 m | Input data |
| Section 3.2 | Number of slices | 4 | Input data |
| Section 3.2 | Segment size | 12 m | This paper |
| Section 3.3 | Initial window side | 4 m | This paper |
| Section 3.3 | Distance between clusters | 0.25 m | Input data |
| Section 3.3 | Distance between cluster means | 0.15 m | Input data |

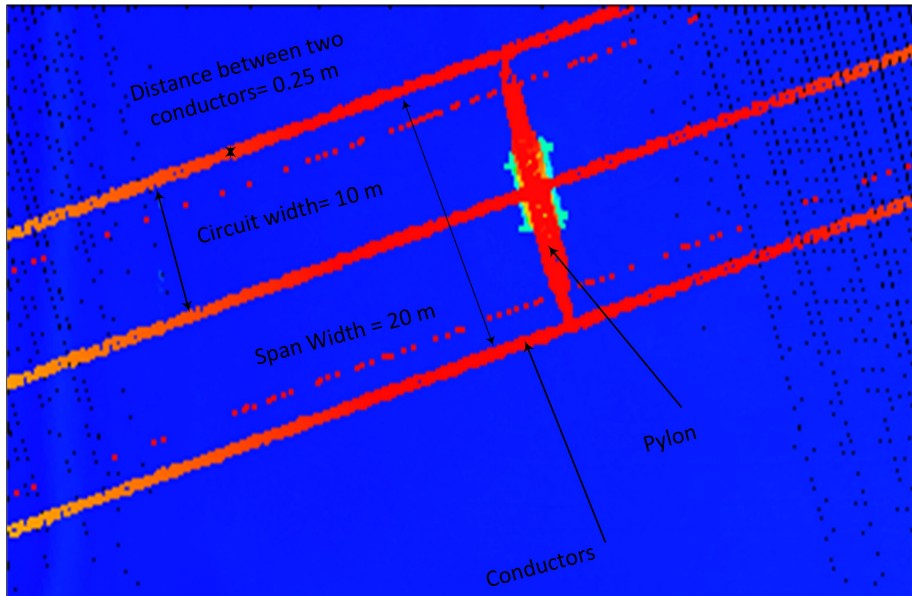

**Figure 9.** Power line infrastructure parameters.

*4.3. Ground Truth*

The evaluation has been carried out in two aspects: object-based and point-based evaluations. In object-based evaluation, the number of subconductors extracted in all bundles and spans is considered. This has been decided by manual inspection of the extracted conductors against the actual conductors in the two data sets. For *n*-conductor bundles, where $n = 1, 2, 4$, the result has been provided separately to analyze whether the proposed method is effective to extract conductors from different types of bundles.

In point-based evaluation, the number of individual conductor points is considered. The points of each extracted conductor are compared to that of the actual conductor. A 3D interface has been developed using MATLAB programming to manually collect the ground truth points. The input points are divided into four object classes: pylons, conductors, vegetation, and ground. As the size of the data sets is very big it is hard to collect the ground truth for the whole data sets. Therefore, only the first three spans from the MDP site and two spans from the BDB site are used for the point-based evaluation of the proposed scheme. Table 4 shows the summary of the ground truth used for evaluation.

Figure 10a,b displays the the ground truth spans from the MDP and BDP sites, respectively. The vegetation and pylon points are not displayed in the ground truth. The left snapshot shows total 5 bundles in three circuits in a span from the MDP site and each bundle has two conductors except the top two which contain single conductors. While the right snapshot is from the BDB site of the first span which has a total of 14 conductors in 2 circuits. It is worth noting that in both the spans the top conductors are thin and have large gaps between points. The ground truth data for the spans which have 4 subconductors in a bundle have not been collected, as the individual conductors are very close and it is hard to separate them manually.

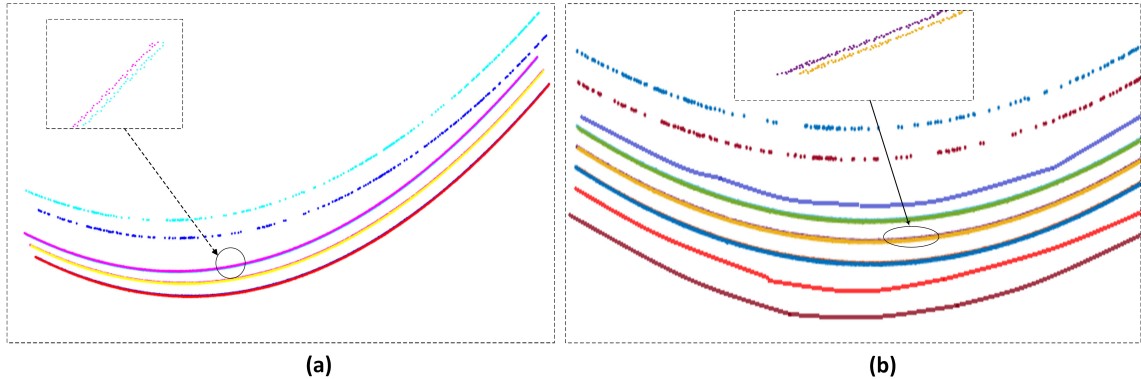

**Figure 10.** Ground truth examples of power line points: (**a**) Maindample (**b**) Bindebango.

**Table 4.** Summary of the ground truth.

| Sites | Areas (m$^2$) | All Points | Conductors Points | Spans | 2-Conductor Bundles | 1-Conductor Bundles |
|---|---|---|---|---|---|---|
| MDP | $1170 \times 330$ | 56,515 | 50,679 | 6 | 18 | 12 |
| BDB | $667 \times 530$ | 435,579 | 36,062 | 6 | 12 | 24 |
| Total | | 492,094 | 86,741 | 12 | 30 | 36 |

### 4.4. Evaluation Metrics

For performance evaluation, the object-based and point-based completeness $C_m$, correctness $C_r$, and quality $Q_l$ metrics are used [27]. For object-based evaluation, the total number of detected pylons, spans and conductors in terms of bundles and in terms of their individual numbers are considered. While for point-based evaluation, the extracted points of pylons and individual conductors are estimated against the ground truth presented above.

## 5. Results

The results for span and pylon extraction are provided in Section 5.1, while the results for extraction of bundle conductors and their object- and point-based evaluations are presented in Section 5.2.

### 5.1. Span and Pylon Extraction

Figure 11 depicts the extracted pylons in the two data sets and Table 5 displays the object-based $C_m$, $C_r$, and $Q_l$ values for the pylons in the two data sets. For object-based evaluation, the total number of extracted pylons and the total number of extracted spans in both data sets are considered. In the MDP data set, all the 26 pylons and the 28 spans, except the 3 poles and 2 spans located in the DLC (due to missing points as mentioned earlier), are extracted. While in the BDB data set all the 24 pylons and 30 spans in all three corridors are extracted.

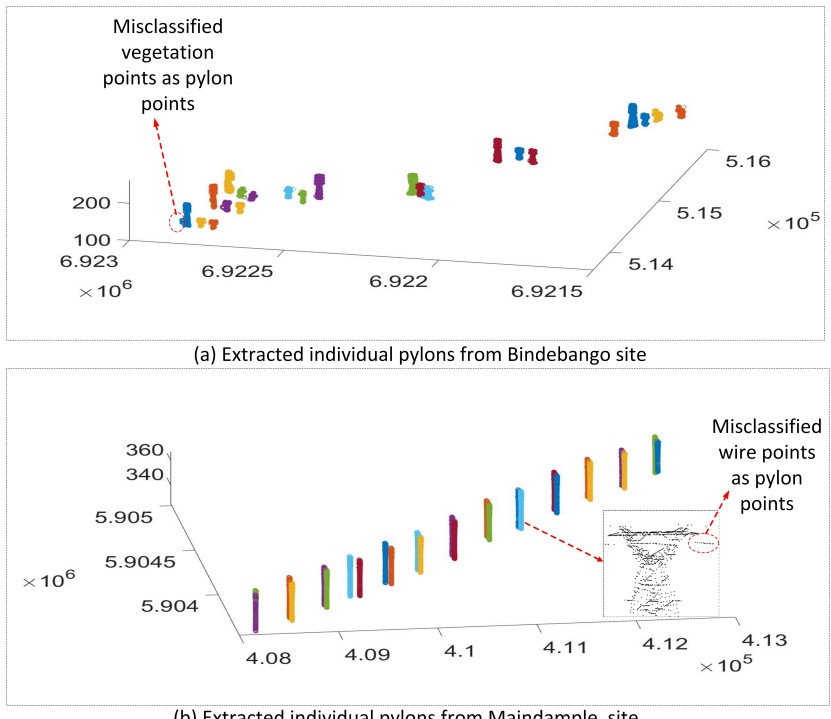

(a) Extracted individual pylons from Bindebango site

(b) Extracted individual pylons from Maindample site

**Figure 11.** Individual pylons extraction in Bindebango (BDB) and Maindample (MDP) sites.

**Table 5.** Object-based evaluation of pylons and spans on the whole test data sets (all values in percentage).

| | Pylons | | | Spans | | |
|---|---|---|---|---|---|---|
| **Data Sets** | **Comp.** | **Corr.** | **Qual.** | **Comp.** | **Corr.** | **Qual.** |
| MDP | 98.2 | 100 | 98 | 90.3 | 100 | 90.3 |
| BDB | 100 | 100 | 100 | 100 | 100 | 100 |
| Average | 99.1 | 100 | 99 | 95.1 | 100 | 95.1 |

### 5.2. Bundle Conductor Extraction

Some bundle and PL extraction results from the MDP and BDB data sets are shown in Figures 12 and 13, respectively, and each subconductor is colored differently to easily distinguish them. Although the subconductors in a bundle reside very close to each other, each subconductor is still well extracted effectively by the proposed method.

For the MDP data set, all the bundle conductors in both corridors are correctly extracted except the ones which are located in the DLC due to shortage of points. All the spans in this site has three 2-conductor bundles and two 1-conductor bundles on the top (see Figure 10). There are no 4-conductor bundles in this site. The extraction results for the MDP data set are shown in Table 6. The object-based $C_m$, $C_r$, and $Q_l$ values are estimated for bundle subconductors and single conductors for each corridor. All 224 conductors in TLC 1 and TLC 2 ($(42 \times 2 + 28) + (42 \times 2 + 28)$) in both corridors of the site are correctly extracted. However, the 18 conductors, i.e., ($6 \times 2 + 6$) exist in the short DLC are missed. Thus, Table 6 shows that TLC 1 and TLC 2 have 100% $C_m$, $C_r$, and $Q_l$, but the overall accuracy and correctness are low due to undetected conductors in the DLC.

**Table 6.** Objected-based evaluation of conductor extraction in Maindample site.

| Corridors | Extracted Conductors | | (Extracted/Total) | Object-Based Evaluation | | |
|---|---|---|---|---|---|---|
| | 2-Conductor Bundles | 1-Conductor Bundles | Total Conductors | Comp. % | Corr. % | Qual. % |
| 1 | 42/42 | 28/28 | 112/112 | 100 | 100 | 100 |
| 2 | 42/42 | 28/28 | 112/112 | 100 | 100 | 100 |
| 3 | 0/6 | 0 /6 | 0/18 | 0 | 0 | 0 |
| Total | 84/88 | 56/62 | 224/238 | 92 | 100 | 92.5 |

The magnified version of Corridor 1, Span 2 in Figure 12 shows that there is no noticeable segmentation error in the MDP site. It is worth noting that single conductors in both spans have sparseness and gaps, but are correctly extracted (see Figure 12b).

As discussed in Section 4.1, in Corridor 1 of the BDB site, the number of conductors in individual spans varies. For example, in each of Spans 1 and 2 there are 14 conductors, i.e., six bundles with two subconductors in each and two single conductors at the top. The subconductors in each bundle are in horizontal arrangement. Each of the other spans in this corridor has 28 conductors, six bundles with 4 subconductors in each, and 2 bundles with 2 subconductors in each. The subconductors in the 4-conductor bundles are in vertical and horizontal arrangement. However, most of the bundles are fully extracted without much segmentation error (Figure 13). Majority of the subconductors including the ones at the top with few points are also fully extracted (Figure 13k,l), though some subconductors are extracted as single conductors due to their alignment issues in the span ( Figure 13i,j). The magnified version in Figure 13j shows that bundle subconductors seem to be appear as single conductors in the middle, thus the last clustering step accumulates them as single conductor points. Figure 13l shows the magnified version of top conductors, it can be noticed that the top (thin) conductors have low point density as compared to the bottom (thick) ones.

Table 7 shows the object-based $C_m$, $C_r$ and $Q_l$ for the extracted conductors from the BDB site against the total number of conductors in Table 2. Most of the individual conductors in all bundles are extracted except those exist in the third span of each corridor. These conductors are not extracted correctly due to noise and error in PL points, some individual conductors seem to intersect each other might be due to wind or acquisition error (Figure 13i,j), thus bundle subconductors in each of these span have been extracted as single conductors. Thus, affecting the $C_m$, $C_r$, and $Q_l$ in all three corridors.

**Table 7.** Objected-based evaluation of conductor extraction at the Bindebango site.

| Corridors | Extracted Conductors | | | (Extracted/Total) | Object-Based Evaluation | | |
|---|---|---|---|---|---|---|---|
| | 4-Conductor Bundles | 2-Conductor Bundles | 1-Conductor Bundles | Total Conductors | Comp. % | Corr. % | Qual. % |
| 1 | 36/42 | 12/12 | 24/24 | 192/216 | 91.4 | 95.0 | 87.2 |
| 2 | 2/2 | 19/21 | 15/15 | 61/65 | 93.8 | 88.4 | 83.5 |
| 3 | 1/1 | 22/25 | 17/17 | 61/65 | 91.5 | 91.6 | 85.5 |
| Total | 39/45 | 53/58 | 56/56 | 318/352 | 92.2 | 90.4 | 86.0 |

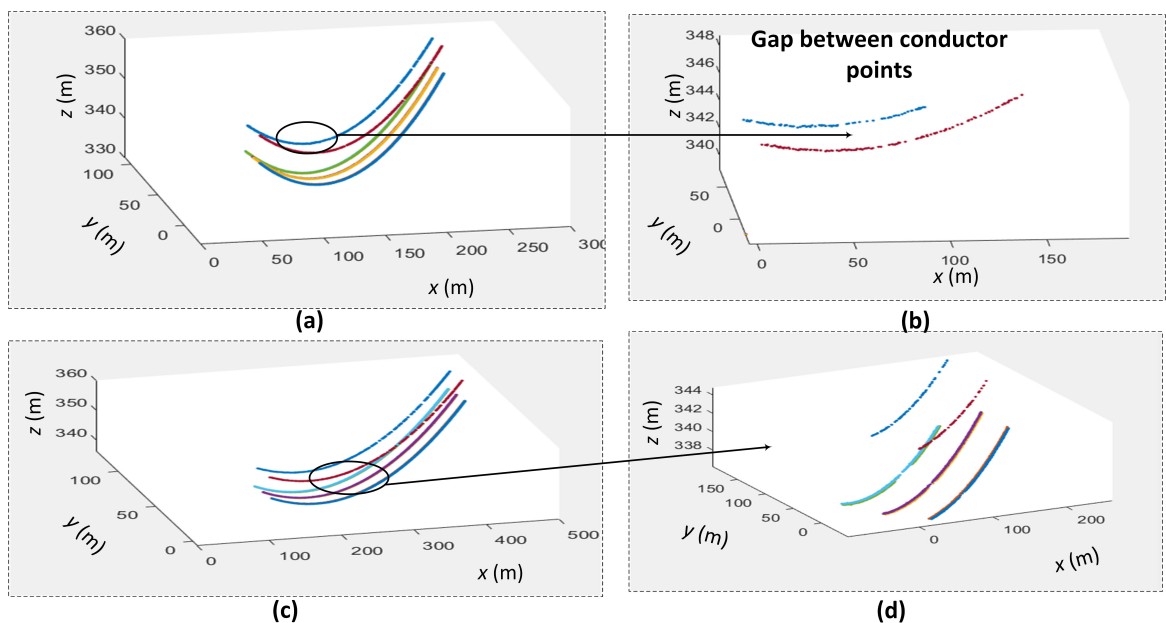

**Figure 12.** Extraction of individual conductors in Maindample (MDP) site. In panel (**a**) the origin (0, 0) represents (4.081 × 10^5, 5.9036 × 10^6), in panel (**b**) the origin (0, 0) represents (4.084 × 10^5, 5.90368 × 10^6), in panel (**c**) the origin (0, 0) represents (4.086 × 10^5, 5.90368 × 10^6) and in panel (**d**) the origin (0, 0) represents (4.087 × 10^5, 5.90368 × 10^6).

In addition to the object-based evaluation, the point-based evaluation is also estimated and its overview is given in Table 8. The extracted pylons and PL points are estimated against the ground truth presented in Section 4.3. For pylons, the completeness and correctness are lower in the BDB site than in the MDP site due to the vegetation points near the pylons being misclassified as pylon points (see Figure 11a). For conductors, the completeness value are low in both data sets as PL points close to pylon points were misclassified as pylons, and vice versa. As discussed in Section 4.1, the ground truth has not been collected for the whole data sets but only the two-tenth of both data sets have been used as the ground truth. Therefore, the results may vary if the point-based results would have been estimated against the complete ground truth of the two data sets.

**Table 8.** Point-based evaluation of pylons and spans on both data sets (all values in percentage).

| Data Sets | Pylons | | | Conductors | | |
|---|---|---|---|---|---|---|
| | Comp. | Corr. | Qual. | Comp. | Corr. | Qual. |
| MDP | 98.2 | 97.2 | 97.1 | 96.3 | 100 | 97.3 |
| BDB | 93.4 | 95.2 | 90.3 | 97.9 | 100 | 96.5 |
| Average | 95.8 | 96.2 | 95.2 | 97.1 | 100 | 96.9 |

Table 9 lists the running time recorded for different steps of the proposed method when it is applied to both data sets. For each step in the table the average time is recorded from about ten runs. The time is proportional to the number of input points. Note that the most time-consuming step is the extraction of individual conductor, which is dependent on the window size. If the density of data set is high, the size can be increased thus the running time will be reduced. It might be noted, parallel computing can be applied to extract conductors from multiple spans simultaneously.

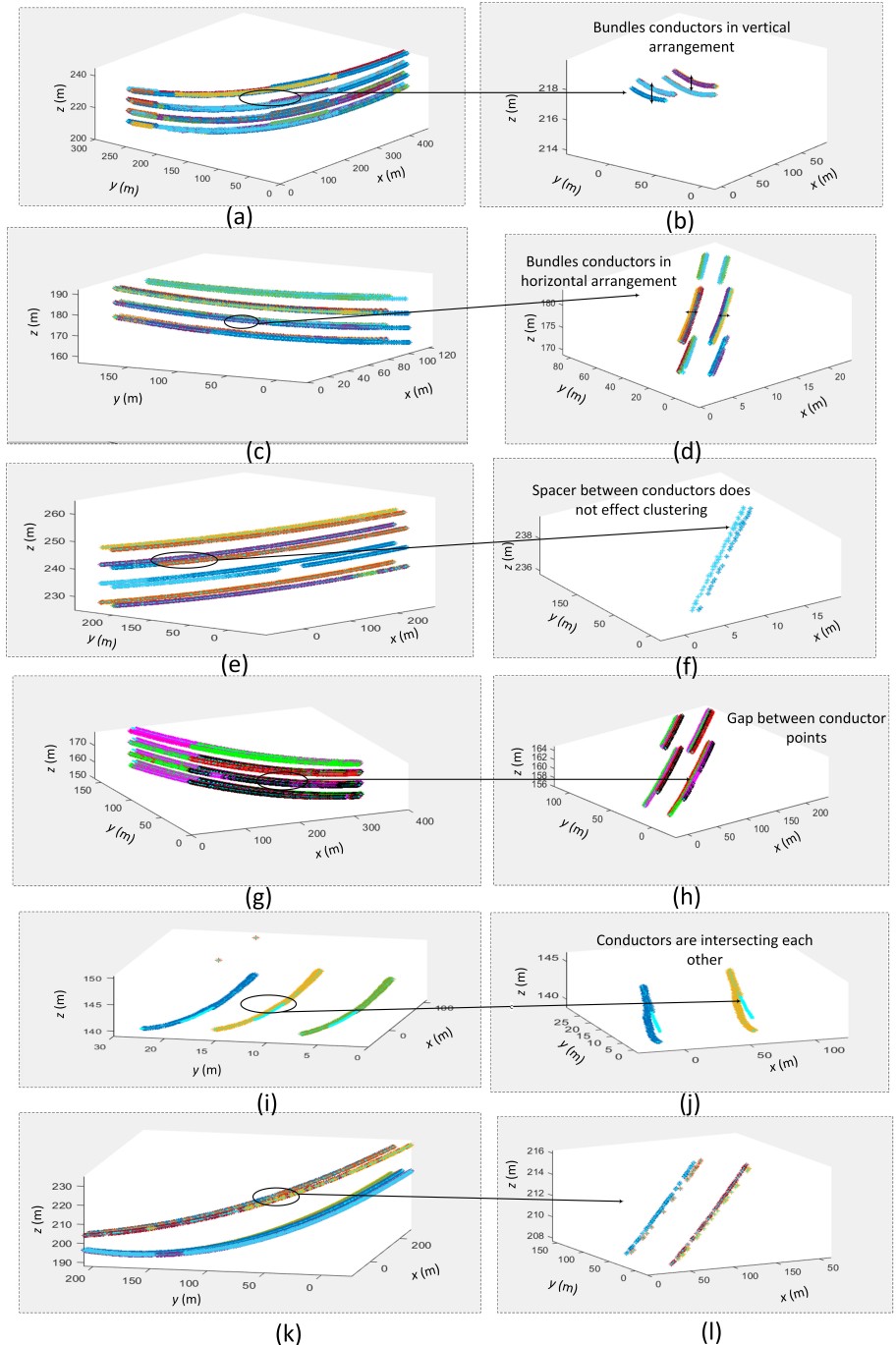

**Figure 13.** Extraction of individual conductors at the Bindebango (BDB) site. In panel (**a**) the origin (0, 0) represents (5.155 × $10^5$, 6.9218 × $10^6$), in panel (**b**) the origin (0, 0) represents (5.159 × $10^5$, 6.9218 × $10^6$), in panel (**c**) the origin (0, 0) represents (5.145 × $10^5$, 6.92255 × $10^6$), in panel (**d**) the origin (0, 0) represents (5.147 × $10^5$, 6.92255 × $10^6$), in panel (**e**) the origin (0, 0) represents (5.1545 × $10^5$, 6.92168 × $10^6$), in panel (**f**) the origin (0, 0) represents (5.1546 × $10^5$, 6.92168 × $10^6$), in panel (**g**) the origin (0, 0) represents (5.147 × $10^5$, 6.92178 × $10^6$), in panel (**h**) the origin (0, 0) represents (5.149 × $10^5$, 6.92178 × $10^6$), in panel (**i**) the origin (0, 0) represents (5.141 × $10^5$, 6.9227 × $10^6$), in panel (**j**) the origin (0, 0) represents (5.143 × $10^5$, 6.9227 × $10^6$), in panel (**k**) the origin (0, 0) represents (5.151 × $10^5$, 5.9219 × $10^6$) and in panel (**l**) the origin (0, 0) represents (5.153 × $10^5$, 5.9219 × $10^6$).

**Table 9.** Processing time (in minutes) for each step.

| Data Set | Corridor | No. of Points in Span | Extraction of One Span | Extraction of One Bundle | Extraction of an Individual Conductor |
|---|---|---|---|---|---|
| MDP | 1 | 18,284 | 3.4 | 3.6 | 5.8 |
| MDP | 2 | 18,275 | 3.4 | 3.7 | 5.7 |
| BDB | 1 | 15,482 | 2.2 | 2.8 | 4.8 |
| BDB | 2 | 12,001 | 1.8 | 2.4 | 3.9 |
| BDB | 3 | 9683 | 1.5 | 2.3 | 3.7 |

## 6. Discussions

As discussed in Section 4.1, the size of the data sets is large and spans in these data sets are different in nature, this give us an opportunity to analyze the effect of some natural factors such as gaps or breakage in power lines and the point density which is low in single conductors and on the proposed method. So, in this section, the proposed method robustness to density and breakage effect on data visually analyzed and discussed. Also, the comparisons with existing method that focused on extraction of bundle conductors using LiDAR data is provided in detail.

### 6.1. Robustness to Data Quality

As mentioned in Section 4.1 the density of both data sets is not high, thus misidentification of conductors can occur when the point density or data quality is low. To visually analyze the impact of the data quality required for the successful bundle conductors extraction, a set of experiments were conducted on spans where the points are sparse and has low density data. As shown in Figure 14, the power lines have few points, but due to sliding window points will be stayed connected to each other. However, errors always occur when points are sparse and their structure is not apparent. For example in span 3 due to noise or power lines structure error the conductors are not individually extracted.

### 6.2. Robustness to Breakage

Sparseness and large gaps often occur when a section of PL is obscured by vegetation or lacks in data points, leading to a small number of power lines being split into several parts, or being undetected. To analyze the robustness to breakage, a set of experiments were conducted by the proposed method. Taking a single conductor as an example, as shown in Figure 15, there was a large breakage due to missing points. This situation will generate non-overlapping windows, thus more clusters will be formed from same power lines, consequently the power lines were divided into several parts. However, the last step of an algorithm will combine the clusters that belong to same conductor, by estimating their mean distances, if it is less then threshold it will be considered as same conductor cluster. The broken fully extracted conductor is shown in Figure 15. It shows that the proposed method could efficiently reduce the impact of breakage.

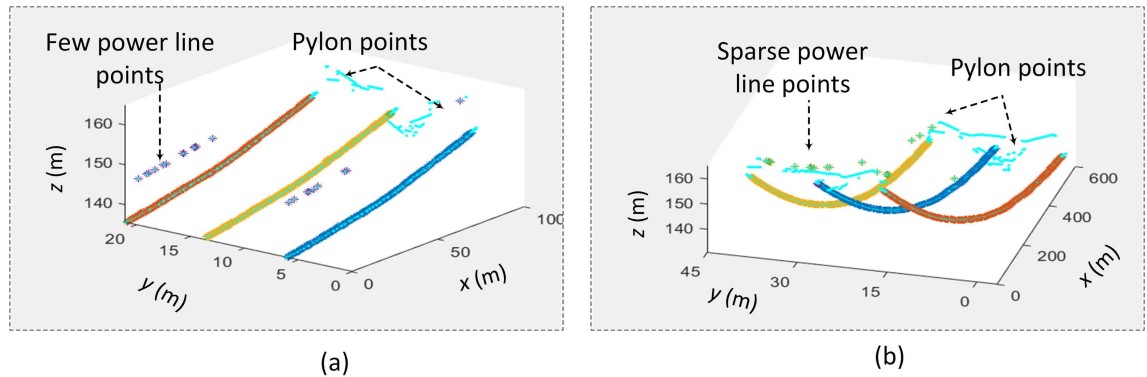

**Figure 14.** Robustness to density. In panel (**a**) the origin (0, 0) represents $(4.086 \times 10^5, 5.90368 \times 10^6)$ and in panel (**b**) the origin (0, 0) represents $(5.136 \times 10^5, 6.92272 \times 10^6)$.

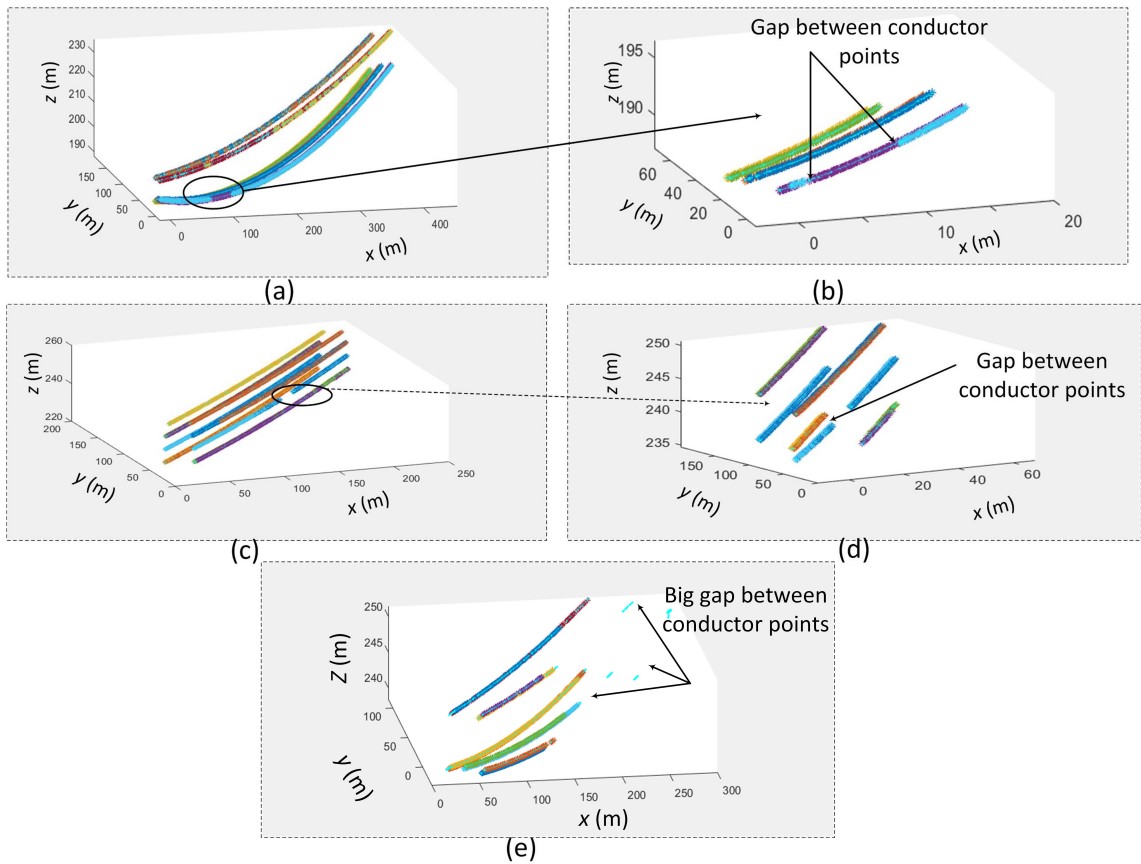

**Figure 15.** Robustness to breakage. In panel (**a**) the origin (0, 0) represents $(5.151 \times 10^5, 6.9219 \times 10^6)$, in panel (**b**) the origin (0, 0) represents $(5.152 \times 10^5, 6.9219 \times 10^6)$, in panel (**c**) the origin (0,0) represents $(5.1545 \times 10^5, 6.92178 \times 10^6)$, in panel (**d**) the origin (0,0) represents $(5.1547 \times 10^5, 6.92178 \times 10^6)$ and in panel (**e**) the origin (0, 0) represents $(5.156 \times 10^5, 6.92164 \times 10^6)$.

### 6.3. Comparison with Existing Methods

As mentioned in Section 1, most of the existing methods paid attention for extraction of single power lines or considered bundle conductors as single. The novelty of this study is automated extraction of bundle conductors on mountainous terrain with comparatively low point density data. Although, it is not fair to do the comparisons as data sets are different. However, the general comparison of existing methods that focused on extraction of bundle conductors is listed in Table 10. The various evaluation methods and performance metrics used in the literature are investigated and listed in Table 10. It can be noticed in Table 10 that many methods depend on supplemental data such as training data and are not fully automated, thus the versatility of the approaches is limited. Nevertheless, our entire approach is fully automated except that it requires the selection of a few parameters. Further, it is important to note here, most approaches show high completeness and correctness in object and point based evaluation metrics, but they typically have been tested on a limited number of spans or they did not extract four bundle conductors. To our knowledge, only one study in the literature has extracted up to four bundle conductors but it has used very high density point cloud data. Thus raises concerns about the extraction of bundle conductors in low point density data. On the other hand, proposed method used very low point density data sets thus have sparseness in PL points.

Despite these advancements, there are still some limitations of the proposed method. It is applicable to high-voltage power lines located in the forest area and the focus of the proposed method is extraction of individual subconductors. In urban areas, where the poles and trees are similar additional steps (postprocessing) will be required to remove trees, buildings, and other objects.

**Table 10.** Comparison of performance and evaluation with existing methods.

| Methods | Data Sets Details | | | | | Bundle Conductors | | Point-Based Evaluation | | | Object-Based Evaluation | | | Supplemental Data |
|---|---|---|---|---|---|---|---|---|---|---|---|---|---|---|
| | No. | Areas (m$^2$) | Points/m$^2$ | Points in Millions | Spans | Actual | Extracted | Comp. % | Corr. % | Qual. % | Comp. % | Corr. % | Qual. % | |
| Jwa et al. [35] | I. | 20,940× 385 | 5 | 1.8 | 7 | 2 | 2 | NA | NA | NA | 93.8 | NA | NA | - |
| Awrangjeb [36] | I. | 5560 × 330 | 23.7 | 32.7 | 26 | 2 | 2 | 95 | 100 | 94.9 | 92.6 | 99.6 | 92.3 | - |
| | II. | 2500 × 430 | 56.4 | 18.5 | 24 | 4 | 2 | | | | | | | |
| Munir et al. [37] | I. | 5560 × 330 | 23.7 | 32.7 | 26 | 2 | 2 | 97.9 | 98.9 | 97.01 | 92.5 | 96 | 92.5 | training data |
| Munir et al. [38] | I. | 1457 × 330 | 23.7 | NA | 10 | 2 | 2 | 99.1 | 100 | 99.01 | 99.05 | 100 | 98.8 | pylons information |
| | II. | 850 × 430 | 56.4 | 18.5 | 6 | 4 | 4 | | | | | | | |
| Zhou et al. [20] | I. | 40 × 2000 | 548.5 | 43.8 | NA | 2 | 2 | NA | NA | NA | NA | 100 | NA | training data |
| | II. | 40 × 730 | 563.5 | 25.2 | | 4 | 4 | | | | | | | |
| | III. | 80 × 600 | 9.6 | 4.5 | | 1 | 1 | | | | | | | |
| Proposed method | I. | 5560 × 330 | 23.7 | 32.7 | 26 | 2 | 2 | 97.1 | 100 | 96.9 | 92 | 95 | 89.5 | - |
| | II. | 2500 × 530 | 56.4 | 18.5 | 24 | 4 | 4 | | | | | | | |

## 7. Conclusions

For automated modeling of PLCs, the extraction of individual power lines is an essential requirement. A robust method is sought that can extract individual power lines on complex power line structures and in intricate environments. Nevertheless, most of the existing research has paid more attention to single PL extraction and limited methods have been reported for bundle subconductor extraction.

Thus, this paper has proposed a novel method for high-voltage power line extraction, which could precisely extract subconductors from the bundle. After the extraction of PL spans, bundles located in each span are detected and extracted using their distribution properties in each span. A sliding window, which moves iteratively over each bundle, is introduced. In each iteration, the points within each window position are projected to a orthogonal plane and recursive clustering procedures are adopted for complete individual conductor extraction.

The proposed method has the following advantages. (1) The structure characteristics of bundle conductors in high-voltage transmission corridors are used for their identification and extraction, and (2) the proposed method can efficiently and precisely extract each subconductor and is robust to noise and breakage. Overall, the proposed method can work in the real structure of power lines. It is worth noting that the extraction of bundle conductors relies highly on the point density and data quality. Errors always occur when points are sparse and their structure is not apparent. In our future work, for better power transmission inspection and management, the power line reconstruction models will be investigated. As the distribution of power line corridors is becoming more complicated (e.g., multi-loop and multi-bundle), more attention will be paid on more applicable and general 3D methods for bundle conductor extraction and modeling.

**Author Contributions:** Conceptualization, N.M. and M.A.; methodology, N.M.; software, N.M.; validation, N.M. and M.A.; formal analysis, M.A.; investigation, N.M.; resources, N.M.; data curation, M.A.; writing—original draft preparation, N.M.; writing—review and editing, M.A.; visualization, M.A.; supervision, M.A and B.S. All authors have read and agreed to the published version of the manuscript.

**Funding:** This research received no external funding.

**Acknowledgments:** The authors would like to express their gratitude to the editors and the reviewers for their constructive and helpful comments for the substantial improvement of this paper. The dataset was provided by the AAM Brisbane QLD Australia.

**Conflicts of Interest:** The authors declare no conflict of interest.

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
