# Peer review of "Automatic Extraction of High-Voltage Bundle Subconductors Using Airborne LiDAR Data"

_remotesensing, doi:10.3390/rs12183078_

Round 1

Reviewer 1 Report

The authors describe a recursive method for automatic extraction of Transmission Lines sub-conductors using airborne LiDAR data, by extending their previously published works. The article is well-written and easy to follow the text.

Some comments about the chapters:

  1. Introduction:
    • The need for the individual separation of conductors is not clear, the motivation of the work is missing, as well as the advantage of doing it (besides more accurate modeling of the PLC);
    • The practical application/objective of the method is not referred.
  2. Related work:
    • Especially in the PL points classification, the focus is on Machine Learning techniques, despite the method's objective of not relying on it. This could be completed with some works that that are not ML-based.
  3. Performance Study:
    • In line 354 is said that the DLC will be excluded from the analysis since it lacks point resolution, however, it is later considered in table 6;
    • Almost all the figures lack referential labels and units, which makes the interpretation of the images harder and confusing. Also, keeping the units with the power of 5 (or 6) while analyzing a small portion of the span is not a good approach (examples: figures 3-6, 12-14). If the referential origin was set to 0 for explanation purposes, you would gain useful graph area for the same image size;
    • Table 8: "Conductors" should be centered, "corr" and "qual" missing capital letter for coherence;
    • Point-based evaluation results can be misleading since they are only considering a portion of the dataset. Especially in the BDB, where it is referred that the span properties vary along the spans;
    • Processing times might be an interesting analysis to make.
  4. Conclusion:
    • Not a comment, but a suggestion. In future work, it may be interesting to also use the vertical catenary conductor model to recover from breakages, which would make the algorithm less dependent on the sliding window size. Some works that explore this:
      • https://ieeexplore.ieee.org/document/1621083
      • https://www.mdpi.com/1424-8220/19/8/1812
      • https://www.sciencedirect.com/science/article/abs/pii/S0924271620300885

Reviewer 2 Report

The paper presents an innovative methodology for the automatic extraction of High-Voltage Bundle Sub-Conductors  using airborne LiDAR Data

The topic is worth of investigating. The quality of the manuscript is excellent, in particular the results presentation and the methodology are high impact.

This reviewer do not find methodologic or scientific reasons that could help the authors improving the quality of their manuscript.

I suggest the editor to accept this paper in the present form.

Reviewer 3 Report

It is a well-written manuscript, a little descriptive in some occasion but it is not a problem. I just only have a few suggestion to be clarified:

  • I recommend to move section 3 (contribution) to Method section.
  • How did you classify ground or vegetation points?
  • What about vegetation? Do you think that can generate noise if vegetation is which shape similar to poles? or maybe so close to power line?
  • How time efficency is your development?

Round 2

Reviewer 1 Report

Thank you for addressing all the previous comments.

Especially making the graphs' origin to 0 made the reading and interpretation much easier.

Adding the processing times was also interesting to have a better idea about the algorithm's processing. Not mandatory, but you can also add a column with the number of points of the span in table 9. This would give an even better perception of the processing dependence on the number of points growth.

Just 2 more remarks:

  1. Duplicated "the large part" in line 56.
  2. Table 9 has MDP corridor 1 repeated.

Congratulations on your work!
